# Static and Dynamic Properties of Al-Mg Alloys Subjected to Hydrostatic Extrusion

**DOI:** 10.3390/ma15031066

**Published:** 2022-01-29

**Authors:** Wojciech Jurczak, Tomasz Trzepieciński, Andrzej Kubit, Wojciech Bochnowski

**Affiliations:** 1Mechanical and Electrical Engineering Department, Polish Naval Academy, 81-103 Gdynia, Poland; w.jurczak@amw.gdynia.pl; 2Department of Manufacturing and Production Engineering, Rzeszow University of Technology, al. Powst. Warszawy 8, 35-959 Rzeszów, Poland; tomtrz@prz.edu.pl; 3Centre for Innovative Technologies, University of Rzeszów, ul. Pigonia 1, 35-959 Rzeszów, Poland; wobochno@ur.edu.pl

**Keywords:** Al-Mg alloy, dynamic strength, fracture morphology, hydrostatic extrusion, work hardening

## Abstract

The aim of this study is to determine the influence of the amount of magnesium in Al-Mg alloys and strain rate on the grain refinement and mechanical properties of the material as determined in a dynamic tensile test. Hydrostatic extrusion was used to process the material. This method is not commonly used to impose severe plastic deformation of Al-Mg alloys. The article presents the results of static and dynamic strength tests on aluminium alloys subjected to plastic deformation in the hydrostatic extrusion process. Technically pure aluminium Al99.5 and three aluminium alloys with different magnesium content, Al-1Mg, Al-3Mg and Al-7.5Mg, were used in the tests. The samples were subjected to static tests using the uniaxial tensile test machine and dynamic tests using a rotary hammer. Compared to pure aluminium, increasing the magnesium content in Al-based alloys strengthened them in hydrostatic extrusion (logarithmic strain ε = 0.86) and caused an increase in the static ultimate tensile stress R_m_, relative strain ε_r_ and the value of the yield stress. For strengthened aluminium alloys, an increase in the strain rate from 750 to 1750 s^−1^ caused an increase in the dynamic ultimate tensile stress from 1.2 to 1.9 times in relation to the static ultimate tensile stress. The increase in magnesium content results in the formation of a larger strengthening phase, influences a different state of stress during dynamic loading and leads to a change in the orientation of the fracture surface. It was also found that an increase in magnesium content is associated with an increased number of voids, which is also directly proportional to the strain rate in the dynamic rotary hammer test.

## 1. Introduction

In the process of hydrostatic extrusion (HE), the workpiece material is surrounded by a liquid medium in the working chamber. Therefore, there is low friction between the product and the container [1]. The moving piston generates pressure in the chamber by compressing the liquid. After reaching the appropriate pressure, the product is extruded through the die. The advantage of the process is the triaxial stress state acting on the workpiece in the working chamber, inhibiting the formation of cracks and their propagation in the deformed material [2,3]. This allows the deformation of hard-to-deform materials due to external stress and extrusion ratios that are higher than other extrusion processes at ambient temperature [1]. The HE process is one of the so-called high plastic deformation—severe plastic deformation (SPD), techniques. The technology of HE allows the grain to be refined to the nanometric level, which results in a significant improvement in the mechanical properties of the material. HE enables the production of nano-grained and ultra-fine-grained structures in metals, such as steel, titanium, aluminium and its alloys, copper and nickel. HE is optimal for fabricating products with complex shapes and a high extrusion ratio because the processing conditions enable the application of uniform pressure on the product [1]. In order to produce difficult-to-deform hexagonal close-packed metallic rods with a high length to diameter ratio, novel SPD hydrostatic cyclic extrusion–compression (HCEC) methods have been developed [4].

Al-Mg alloys, thanks to their low density and high strength as well as very good corrosion resistance in aggressive environments, are used as a construction material, in particular, in the shipbuilding and aircraft industries. In order to provide them with sufficient high strength properties, they are stabilised by adding appropriate alloying additives, such as manganese, chromium or titanium in an amount not exceeding 3%. The effect of this treatment is that the material has a finer grain, better corrosion resistance, higher stability at elevated temperatures and increased strength properties. Copper improves mechanical properties and enables precipitation strengthening. Iron strengthens pure aluminium. However, it is considered an impurity because it creates very brittle needles of the δ-Al_4_Si_2_Fe phase. Titanium promotes grain refinement in the casting process [5].

Aluminium metal matrix composites (AMMCs) fabricated using powder metallurgy are expanding their range due to their characteristics, such as high-temperature resistance, high specific strength and lightweight than conventional materials. Behera et al. [6] fabricated an aluminium–metal–matrix-composite (Al–0.5Si–0.5Mg–2.5Cu–5SiC). They found that the incorporation of fine SiC particles into sintered matrix elements can improve erosive wear resistance by a factor of 200–300%. From the observations of many authors (Abbas et al. [7], Behera et al. [8], El-Aziz et al. [9]), it is noted that AMMCs are capable of improving the erosion behaviours and degradation of corrosion. Behera et al. [10] found that the hardness of sintered AMMC is increased with an increased number of Si-C reinforcements.

Several studies have been devoted to the production of ultrafine-grained (UFG) Al-Mg alloys. Lee et al. [11] studied ultrafine-grained Al-Mg7.5 in order to investigate its mechanical behaviour as extruded specimens. Transmission electron microscope (TEM) observation revealed that the average grain size remained 300 nm in the sub-micron level. Kim et al. [4] found that the hydrostatic extrusion of AlMg7.5 aluminium alloy reduced porosity and produced fibrous structure along the extrusion direction [12]. The hydrostatic extrusion consolidated the powder metal to a near void-free state and produced a fibrous inner structure by elongating the coarse-grained clusters along the extrusion direction. Skiba et al. [5] analysed the effect of the reduction in cutting forces of AA5085 aluminium alloy due to strain hardening of the material by HE. The Al-Mg alloy tested after cold cumulative HE was characterised by significant refining of the microstructure to a nanometric scale, which is responsible for a significant increase in yield stress and ultimate tensile stress compared to the undeformed material. Chrominski et al. [13] investigated the effect of the ageing temperature on strengthening mechanisms in UFG Al-Mg-Si alloy processed by HE. Various grain types have been found in HEed aluminium alloy that are the result of different stress conditions during SPD. The substructure present in the particular grain types is responsible for the strengthening mechanism. In another paper, Chrominski et al. [14] studied precipitation strengthened Al-Mg-Si alloy processed by HE. It has been demonstrated that the microstructure after hydrostatic extrusion consists of two types of grain: (i) micron-sized with a dislocation substructure and (ii) nano-sized free of dislocations and surrounded with high angle grain boundaries. Majchrowicz et al. [15] enhanced strength and electrical conductivity of ultrafine-grained Al-Mg-Si alloy processed by HE. They found that higher applied strains during the HE process decreased the age-hardening response of the Al-Mg-Si alloy but accelerate the precipitation kinetics.

As one of the main objectives of this work, the hydrostatic extrusion method should be separately described in more detail. This technique engages hydrostatic pressure for the extrusion of material in the form of a rod through a die, resulting in a simultaneous reduction in the rod diameter and plastic deformation of the material [2,16,17]. Such a triaxial state of stresses results in some exceptional features of this method, such as the homogeneity of deformation, high strain rates in a single pass and low friction, which allows the processing of even brittle and hard-to-deform materials [16,18,19]. It has also been noted that near adiabatic conditions maintained during hydrostatic extrusion result in the generation of heat; therefore, thermally activated processes, such as recovery and crystallisation, may occur [20]. Hydrostatic extruded materials tend to reveal lamellar microstructure with the grains elongated along the deformation direction and an increased contribution of high angle grain boundaries [21].

Most published studies investigated the effectiveness of the plastic deformation methods in improving the strength of Al-Mg alloys. Works were focused on the correlation between the amount of magnesium, deformation degree and strengthening of the processed materials. Many of them also indicated that large plastic deformations resulted in an improvement in properties, such as microhardness, without deteriorating other properties, such as electrical conductivity and ductility. The aim of this study is to determine the influence of the amount of magnesium and strain rate on the grain refinement and mechanical properties of material determined in the dynamic tensile test. Moreover, hydrostatic extrusion was used to process the material in this paper. This process requires special equipment and is not commonly used to impose severe plastic deformation of Al-Mg alloys. In this paper, Al-Mg alloys were strengthened using the HE process and examined through static and dynamic tests. The samples were subjected to static tests using the uniaxial tensile test machine and dynamic tests using a rotary hammer. The effect of strain rate and the composition of magnesium on the fracture behaviour was studied using fractographic analysis and energy-dispersive X-ray spectroscopy. This paper is organised as follows. The research methodologies adopted to characterise and assess the properties of the Al-Mg alloys are presented in Section 2. The results of the static and dynamic tensile tests are presented in Section 3.1 and Section 3.2, respectively. Analysis of the fracture morphology of samples after the dynamic test is shown in Section 3.3. Towards the end of this article, analysis of voids on the fracture surfaces (Section 3.4) and sources of the nucleation of voids (Section 3.5) are discussed. Finally, conclusions and a future work plan (Section 4) are made.

## 2. Materials and Methods

### 2.1. Material

The process of hydrostatic extrusion was applied to technically pure Al99.5 aluminium and three aluminium alloys with different magnesium content, i.e., Al-1Mg, Al-3Mg and Al-7.5Mg. Specimens in the form of rods with a diameter of 20 mm were fabricated by casting and subsequently homogenised. These materials were produced by the Institute of Non-Ferrous Metals (Skawina, Poland), and the detailed manner of their fabrication is the secret of the Institute. The chemical composition of the test materials determined using an optical emission spectrometer is listed in Table 1, Table 2, Table 3 and Table 4.

### 2.2. Hydrostatic Extrusion

The HE process was used to impose plastic deformation on the materials examined. It was performed using a hydrostatic extrusion press (Figure 1) designed and constructed at the Institute of High Pressure Physics UNIPRESS (Warsaw, Poland). The press operates at room temperature and imposes pressures up to 2.5 GPa [18]. The extrusion tool consisted of a die and ram placed in a container (Figure 2).

In the extrusion process, the diameter of the rods was reduced from 20 mm to 13 mm (Figure 3). The accumulated logarithmic (true) strain ε was calculated using Equation (1) [16,22]:(1)$$u=2\ln\frac{\phi_{i}}{\phi_{f}}$$
where $\phi_{i}$ is the initial diameter and $\phi_{f}$ is the final diameter (after deformation).

For brevity, the materials examined that were not subjected to HE will be referred to as IS (initial state). The logarithmic strain for the reduction in the initial diameter of 20 mm into a final diameter of 13 mm is u = 0.86, according to Equation (1).

### 2.3. Static Tensile Test

The basic mechanical properties of the materials tested have been determined using the uniaxial tensile test according to the EN ISO 6892-1:2016-09 standard [23]. Measurements were carried out at room temperature at a strain rate of 0.0027 s^−1^ on an MTS 810 uniaxial servo-hydraulic machine. Three specimens (Figure 4) were tested for each material and the average Basic Mechanical parameters were determined.

### 2.4. Dynamic Tensile Test

An instrumented rotary hammer (Figure 5a) was used for dynamic stretching of the samples (Figure 5b). The measuring equipment of the rotary hammer consisted of a strain gauge dynamometer, a system amplifier, and a computer system for measuring the dynamic parameters. Before the measurements, the indications of the rotational speed of the drum on which the hammer claw was placed were calibrated with the strain gauge dynamometer. Samples for dynamic tests (Figure 6) were prepared on the basis of samples subjected to HE. The tests were carried out for three impact velocities: 15, 25 and 35 m·s^−1^, which corresponds to the following strain rates: 750, 1250 and 1750 s^−1^. Three specimens were tested for each material.

### 2.5. Surface Characteristics

Microstructural examination of the RFSSW specimens was carried out using a Nikon Epiphot 300 light microscope with NIS-Elements V2.3 software. The morphology of the fracture surfaces was examined using a Hitachi S-3400N Scanning Electron Microscope (SEM) (Hitachi, Chiyoda, Japan). The Energy-Dispersive X-ray spectra of specimens were determined using a Hitachi S-3400N microscope equipped with a Thermo Scientific Ultra Dry EDS Detector.

## 3. Results

### 3.1. Static Test

Figure 7a shows the stress–elongation curves obtained for the Al99.5 aluminium and aluminium alloys tested. The basic mechanical parameters for Al99.5 aluminium were as follows: ultimate tensile stress (UTS) R_m_ = 105 MPa, yield stress (YS) R_p0.2_ = 65 MPa and relative elongation ε = 21%. The addition of only 1% Mg (Al-1Mg) caused the appearance of the Portevin–Le Chatelier (PLC) effect, which describes a serrated stress–strain curve above the yield point. A change in mechanical properties of the Al-1Mg alloy in relation to Al99.5 aluminium led to following parameters: R_m_ = 118 MPa (12% increase), R_p0.2_ = 43 MPa (34% decrease) and relative elongation ε = 14% (34% decrease). The occurrence of slight strain hardening and an increase in R_m_-value by 15 MPa caused a reduction in the yield stress by 22 MPa and thus an increase in the plasticity margin by 77% with a clear reduction in plasticity to 13.8%. Increasing the Mg content to 3% in Al-3Mg alloy caused a further increase (in relation to Al99.5 aluminium) in all the parameters describing mechanical properties: R_m_ increased by 90% (to 200 MPa), R_p0.2_ increased by 6.2% (to 73 MPa) and the relative elongation ε increased by 40% (to 29.5%). The PLC effect persisted but with an increased amplitude. This amplitude reduced its range to 7.5% for an alloy with increased Mg content (Al-7.5Mg). The differences in the mechanical properties of Al-7.5Mg alloy in relation to Al99.5 aluminium were as follows: R_m_ increased by 216% (to 332 MPa), R_p0.2_ increased by 115% (to 140 MPa) and relative elongation ε increased by 33% (to 27.9%). The changes in the mechanical properties of the Al-7.5Mg alloy in relation to the Al-3Mg alloy (Mg content increased by 4.5%) were as follows: R_m_ increased by 66% (to 332 MPa), R_p0.2_ increased by 92% (to 140 MPa) and the relative elongation ε decreased by 7% (to 27.9%). The increase in strength of the aluminium alloy with increasing Mg content is confirmed by data found in the literature [16,24].

Figure 7b shows the stress–elongation curves for the materials after HE. Hardening of Al-Mg alloys by HE for the logarithmic strain u = 0.86 increased the strength of these materials while significantly reducing the relative deformation for the same materials which were not subjected HE processing. A change was observed in the course of the tensile curve and the value of the yield strength R_p0.2_. For Al99.5 aluminium, there was an increase in the R_m_ value by 45.7% (to 153.8 MPa), R_p0.2_ increased by 85.6% (to 120.62 MPa) and the relative elongation decreased by 95% (to 1%). For the Al-1Mg alloy in relation to the unhydroextruded material, R_m_ increased by 80.5% (to 213.42 MPa), R_p0.2_ increased by 124% (to 196.34 MPa) and the relative elongation decreased to 85.7% (to 2%). Increasing the magnesium content to 3% in the aluminium rod caused the R_m_ value to decrease by 53.4% (to 306.8 MPa), R_p0.2_ increased by 277% (to 275.51 MPa), but the relative elongation decreased by 92.2% (to 2.3%). The amplitude of the serration increased for the Al-Mg alloy with a magnesium content of 7.5%. For the Al-7.5Mg aluminium alloy in relation to the unhardened material, R_m_ increased by 43.8% (to 477.3 MPa), R_p0.2_ increased by 314% (to 361.57 MPa) and the value of relative elongation decreased by 58.8% (to 11.5%).

In summary, the technology of HE of Al99.5 aluminium and Al-Mg alloy rods increased their strength properties with a significant reduction in their plastic properties. The change in the nature of the tensile curve of the HEed samples in relation to the unprocessed samples substantially changed the value of the yield stress, resulting in a reduction in the plasticity margin. A delayed serration phenomenon for the HEed specimens could also be seen.

After calculating the nominal characteristic σ_nom_ = f(ε_nom_) and the values of R_p0.2_ and R_m_, attention should be paid to the error, especially that which is visible in the plastic range. This results from a change in the initial cross-section of sample A_0_. During stretching, the cross-section decreases and is a function of the load A = A (F). Before the yield point is exceeded (elastic range), these changes are ignored. When the yield point is exceeded, the influence of the change in the sample cross-sectional area on the material characteristics increases until it breaks [25]. This problem also refers to the strains since the current increment is generated at the current length l(F), not at the initial l_0_. The difference between the nominal and true tensile characteristics of a material is shown in Figure 8.

Carrying out engineering calculations concerning the elastic–plastic range of the test material requires the preparation of the parameter σ_true_ = f(ε_true_). The nominal characteristics σ_nom_ = f(ε_nom_) resulting from the assumption of the initial cross-section A_0_ = const. for the entire characteristic makes it impossible to use it in numerical analyses of material forming processes. It was obvious that the cross-section of the specimen was reduced during the uniaxial tensile test. In the elastic range, the changes in the cross-section that occurred were not taken into account due to slight changes. They gained significance when the yield point was exceeded, where the speed of change in the cross-sectional area of the specimen increased. The true stresses are defined by the ratio of tensile force F and the cross-sectional area of the sample depending on this force at a given moment. At the moment of fracture, the measurement of the cross-section is troublesome to perform and only modern techniques (e.g., Digital Image Correlation—DIC) are able to measure the true reduction in the area at fracture. Figure 9 shows selected tensile curves of the aluminium and Al-Mg alloys tested, which were produced using Aramis, a non-contact and material-independent measuring system based on DIC.

The differences in the values of nominal and true tensile stress for individual materials after exceeding the yield point are clearly visible (Figure 9). The higher the magnesium content in the Al-Mg alloys, the greater the difference between the true and nominal ultimate tensile stress R_m_ at similar deformation. These differences can also be demonstrated analytically, taking into account physical and geometric factors. The elementary true strain dε_true_ is defined by the relationship [27]:(2)$${d\varepsilon }_{\mathrm{true}}=\frac{\mathrm{dl}}{l}$$

Thus
(3)$$\varepsilon_{\mathrm{true}}=\int_{l_{0}}^{l}\frac{1}{l}\mathrm{dl}=\ln\left(\frac{l}{l_{0}}\right)$$

True stress σ_true_ is defined as a ratio of tensile force F and the cross-section of the sample at a given moment A(F)
(4)$$\sigma_{\mathrm{true}}=\frac{F}{A\left(F\right)}$$

The mutual relation between the σ_true_ and σ_nom_ is obtained by assuming that the volume of the stretched sample is constant throughout the process, so l_0_A_0_ = lA(F), therefore:(5)$$A\left(F\right)=A_{0}\times \frac{l_{0}}{l}$$
where A_0_ and l_0_ are the initial cross-section of the sample and initial length of the sample measurement zone, respectively.

Therefore:(6)$$\sigma_{\mathrm{true}}=\frac{F}{F\left(A\right)}=\frac{F}{A_{0}}\frac{l}{l_{0}}=\sigma_{\mathrm{nom}}\left(\frac{l}{l_{0}}\right)$$

Nominal strain ε_nom_ can be determined using equation
(7)$$\varepsilon_{\mathrm{nom}}=\frac{l-l_{0}}{l_{0}}$$

Equation (7) can be rewritten as
(8)$$\frac{l}{l_{0}}=1+\varepsilon_{\mathrm{nom}}$$

After substituting Equation (8) in (6) we obtain
(9)$$\sigma_{\mathrm{true}}=\sigma_{\mathrm{nom}}\times \left(1+\varepsilon_{\mathrm{nom}}\right)$$

It is possible to divide the true strain ε_true_ into two parts: elastic ε_sp_ and plastic ε_pl_. The first of these is determined on the basis of Hook’s Law using Young’s modulus E (Figure 10). Plastic deformation can be determined according to Equation (10) [28,29]:(10)$$\varepsilon_{\mathrm{pl}}=\varepsilon_{\mathrm{true}}-\varepsilon_{\mathrm{sp}}=\varepsilon_{\mathrm{true}}-\frac{\sigma_{\mathrm{true}}}{E}$$

Figure 10 also illustrates the range of plastic deformation. Let the sample be loaded with a stress corresponding to point A. As a result of exceeding the yield stress R_e_, the material is not able to return to the starting point after unloading, but only to point B. This proves the formation of a plastic strain ε_pl_. Using the parameters presented in Table 5, it is possible to present the true and nominal characteristics of the material tested.

The mechanical properties of the aluminium and its alloys that were tested were described on the basis of the true characteristics obtained analytically by means of a static uniaxial tensile test. For materials with HE treatment with u = 0.86 (bars with ϕ = 13 mm), an increase in Mg content in the Al-Mg alloys caused an increase in material strength and a reduction in deformation (Table 6). Table 6 also summarises the results of the true mechanical properties of the materials tested and differences in the UTS between the data from the nominal (theoretical) characteristics and the UTS determined from the true characteristics obtained analytically, and by means of Digital Image Correlation. This information is important for the design of structures made of these materials which are subjected to dynamic loading during operation.

The differences in the values of ultimate tensile stresses R_mr_ – R_mn_ determined for the materials tested (the last two columns in Table 6), did not differ significantly with regard to the method of determining these true characteristics. Therefore, in the absence of the expensive Aramis Adjustable instrumentation intended for the deformation and displacement measurements using the DIC technique, the analytical method can be an effective method to use.

### 3.2. Dynamic Test

During the dynamic load of the structure, the deformation speed has a significant impact on the strength properties of the materials. The rate of growth of the strain significantly affects the value of the yield stress. The strain rate is defined as the strain increment over time. The averaged change in sample dimensions is given by the formula:(11)$$\varepsilon =\frac{\Delta l}{l}=\frac{v\;\times \;t}{l}$$

In Equation (11), v defines the speed at which a sample with length l (Figure 11) is stretched in time t, so the strain rate can be represented as follows:(12)$$\dot{\varepsilon }=\frac{d\varepsilon }{\mathrm{dt}}=\frac{d}{\mathrm{dt}}\left(\frac{v\times \;t}{l}\right)=\frac{v}{l}$$

The dynamic tests of samples taken from bars subjected to HE (u = 0.86) were carried out at a minimum of three speeds of the rotary hammer of 15, 25 and 35 m·s^−1^, which corresponds to the following strain rates 750, 1250, 1750 s^−1^. During the dynamic tests, the course of dynamic force versus displacement was recorded. Figure 12 shows an example of the diagram of the dynamic tensile force recorded on the oscilloscope of a rotary hammer and the method of reading the maximum dynamic force adopted for the calculation of dynamic tensile stress R_md_.

An important factor in dynamic tests is the occurrence of differences in heat generation during low- and high-velocity plastic deformation. The majority of the heat generated under quasi-static deformation is conducted and/or convected from areas subjected to this type of deformation. In such conditions, the workpiece remains isothermal. For abrupt deformations, the process is defined as adiabatic, as the short duration of deformation is insufficient to distribute the heat generated [30,31].

For each kind of material tested, three samples were tested at each strain rate. Average values of the dynamic strength R_md_ and dynamic elongation A_5d_ are shown in Figure 13a,b, respectively. For comparison purposes, Figure 13 also includes the values of static properties (columns 4 and 5 from Table 4) for έ~0 s^−1^.

For individual test materials, a comparison was made of the effect of the strain rate on the value of the dynamic strength in relation to the ultimate tensile stress determined in the static test.

In the case of HEed (u = 0.86) Al99.5 aluminium, the following relations were observed:An increase in the R_md_-value 128.5 MPa to 190 MPa at έ = 750 s^−1^ (15 m·s^−1^);An increase in the R_md_-value 128.5 MPa to 250 MPa at έ = 1250 s^−1^ (25 m·s^−1^);An increase in the R_md_-value 128.5 MPa to 313 MPa at έ = 1750 s^−1^ (35 m·s^−1^).

In the case of HEed (u = 0.86) Al-1Mg aluminium alloy, the following relations were observed:An increase in the R_md_-value 191 MPa to 330 MPa at έ = 750 s^−1^;An increase in the R_md_-value 191 MPa to 234 MPa at έ = 1250 s^−1^.

In the case of HEed (u = 0.86) Al-3Mg aluminium alloy, the following relations were observed:An increase in the R_md_-value 327 MPa to 495 MPa at έ = 750 s^−1^;An increase in the R_md_-value 327 MPa to 496 MPa at έ = 1250 s^−1^;An increase in the R_md_-value 327 MPa to 485 MPa at έ = 1750 s^−1^.

In the case of HEed (u = 0.86) Al-7.5Mg aluminium alloy, the following relations were observed:An increase in the R_md_-value 528 MPa to 720 MPa at έ = 750 s^−1^;An increase in the R_md_-value 528 MPa to 731 MPa at έ = 1250 s^−1^;An increase in the R_md_-value 528 MPa to 757 MPa at έ = 1750 s^−1^.

Detailed results of the dynamic tests are included in the content report [31].

For hydroextruded aluminium alloys, the increase in the strain rate from 750 to 1750 s^−1^ caused an increase in dynamic strength R_md_ from 1.2 to 1.9 times when compared to the static strength. A noticeable increase in R_md_ for all the test materials only occurred up to strain rate 750 s^−1^, followed by the stabilisation of R_md_, especially for Al-Mg alloys containing 3% and 7.5% Mg. AlMg1 alloy behaved differently. The value of R_md_ of this alloy decreased after exceeding the strain rate of 750 s^−1^ without exceeding the value of the static strength. Only for Al99.5 aluminium was there an increase in R_md_ and dynamic relative strain ε_d_ to the strain rate of 750 s^−1^. For aluminium alloys hardened through HE, the R_md_-value and ε_d_-value increased, even by as much as a factor of more than five in relation to the static values of the corresponding parameters.

### 3.3. Fracture Morphology of Samples after Dynamic Test

The following tests were performed: SEM fractographic analysis, quantitative analysis of the fracture surfaces, macro photographs of the necking area of the samples and, in the case of Al-1Mg and Al-3Mg samples, an analysis of the chemical composition in selected areas of the fractures. Table 3, Table 4, Table 5 and Table 6 present the view of the tensile specimens after testing and the morphology of fracture surfaces.

During deformation, all specimens break in the ductile fracture mode according to the classical mechanism of (i) void nucleation, (ii) the growth of voids as a result of attaching new dislocations to them and (iii) the coalescence of the voids leading to crack propagation.

On the micro-level, the fracture surfaces of ductile materials exhibit voids generated by dislocation activities at the final fracture stage [32,33]. The nucleation of the voids takes place around the particles at the interface between the reinforcing phase and the matrix. In general, the influence of precipitates on the nucleation of voids increases with the increase in the strain rate at the matrix–precipitation interface. Creating a void is easier when the sizes of the precipitates are larger when the particles take a spherical shape. On selected fractures, intermetallic phases can be locally observed at the bottom of the voids. Depending on the magnesium content, the particles have different sizes and shapes.

In the case of the samples of Al99.5 aluminium, ideal ductile cracking could be observed causing the formation of a characteristic conical reduction in the area at fracture (Table 7, Table 8, Table 9 and Table 10). It was only in the case of the Al99.5 aluminium samples that the cracking proceeded with the formation of a double saucer-shaped fracture characteristic of materials with high plasticity. No saucer-shaped cracking was observed in the remaining samples.

Increasing magnesium content resulted in the formation of a greater strengthening phase and influenced different states of stress during dynamic loading, leading to a change in the orientation of the fracture surface (Table 7, Table 8, Table 9 and Table 10). In such conditions, the share of shear forces increased, leading to the formation of the fracture surface at an angle of approx. 45° to the sample axis. As a result of the increased share of shear stresses, the voids on the fracture surface took a parabolic shape.

The ductility of aged aluminium alloys is strictly dependent on the size, distribution, shape (spherical, disc, rod, needles) and share of the strengthening phase. In the test alloys, there were not only precipitates on a nanometric scale but also particles with a size of several micrometres. When there was the simultaneous occurrence of several types of precipitates/inclusions, the strength and ductility of the alloy changed according to the Pythagorean addition law taking into account the strengthening effect of multiple precipitates.

On the one hand, the strain hardening of the aluminium alloy is determined by the shear modulus of the precipitates, which are obstacles to the movement of dislocations moving in the alloy matrix. On the other hand, the factor controlling the amount of strengthening is the mutual crystallographic orientation of the particle and the matrix. This orientation is beneficial when the dislocation slip plane in the separated particle is not parallel to that in the matrix (then dislocations sliding in the matrix are not able to cut the precipitates). The size, distribution and shape of precipitates were strongly dependent on the chemical composition of the alloy and the microstructure of the material before heat treatment (HT) and on the parameters of the HT.

### 3.4. Analysis of Voids

Depending on the chemical composition of the aluminium alloys tested and the strain rate, the average number of voids and their histogram was related to the projection of the change in the surface area of voids (Table 11, Table 12, Table 13 and Table 14). A quantitative analysis of the voids on the fracture surface was performed for a fracture area of 0.135 mm × 0.118 mm.

Table 15 summarises the quantitative analysis of voids on the fracture surfaces for selected variants of dynamically stretched samples. It can be observed that the increase in the content of the magnesium was associated with an increased number of voids, which number was also directly proportional to the strain rate in the dynamic tensile test. The greater number of voids with a higher content of magnesium was related to the greater number of boundaries between the matrix material and the alloying additive in the material microstructure. This in turn manifests itself in a greater number of nucleated voids under sample loading.

### 3.5. Source of the Nucleation of Voids

The fracture surfaces obtained in the dynamic tensile test were characterised by the presence of voids of various sizes depending on the content of magnesium and the strain rate. These voids testify to the plastic nature of the cracking. The void nucleation phenomenon resulted from the presence of inclusions and particles of various phases in the material. The process of micro-void formation can take place through decohesion at the matrix-particle interface or as a result of the cracking of the particles of the second phase.

In the material subjected to tensile stress, the existing micro-voids undergo an increase which occurs through elastic and plastic deformation of the matrix material. After exceeding a specific value of stress, the voids merge. This phenomenon occurs in an avalanche, which in turn leads to destruction of the material [34].

In the case of Al99.5 samples, perfect ductile cracking was observed, causing the formation of a characteristic narrowing cone at the fracture site. Only in the case of Al99.5 samples did the cracking proceeded with the formation of a double-saucer-shaped fracture characteristic of materials with high plasticity. No cracking of this type was observed in the remaining samples.

In the case of samples containing magnesium, heterogeneity in the structure of the fracture was observed, which manifested itself in a different size and shape of the voids. The shape of the voids, which depends on the local loading conditions, showed a majority of normal stresses. This type of failure may be due to variations in the chemical composition of the material. Figure 14 summarises the results of the EDS analysis for side surfaces of the bars of the basic sample (Figure 14a) and samples with magnesium content (Figure 14b–d), which confirm the local significant variation in chemical composition.

A number of analytical models for the description of nucleation and void growth can be found in the literature [35,36]. Rice and Tracey [37] analysed the growth of voids in the tensile test. A single void was considered with an area *S_v_* and radii in the main directions R_1_, R_2_ and R_3_. Assuming that the void is very small compared to the dimensions of the incompressible elastic–plastic object in which it is located, and this object is influenced by a uniform deformation velocity field $\dot{\varepsilon_{\mathrm{ij}}}$, the speed of growth of the spherical void with radius R in one of the main directions is
(13)$$\frac{{\dot{R}}_{l}}{R_{l}}=A_{1}+{\dot{\varepsilon }}_{l}+A_{2}\dot{\bar{\varepsilon }}$$
where: $\dot{\varepsilon_{\mathrm{ij}}}$ is the sample deformation speed, l is number of the main direction, A_1_ is parameter, taking into account the increased velocity of the void enlargement in the direction of the strain velocity field applied and the strengthening properties of the material surrounding the void, $A_{2}$ is a function of hydrostatic stress, and ${\dot{\bar{\varepsilon }}}_{\mathrm{ij}}=\sqrt{\frac{2}{3}\dot{\varepsilon_{ij}}\dot{\varepsilon_{ij}}}$.

Due to the fact that the nature of ductile fracture related to the nucleation, expansion and connection of voids is significantly influenced by the kind of inclusions and precipitates present in the material microstructure, fractographic analyses were carried out in order to identify the microstructure components of selected variants of the alloys. This analysis was intended to identify hypothetical sources of void nucleation.

The EDX analysis of the chemical composition was carried out on the fractures of samples made of Al-1Mg, Al-3Mg and Al-7.5Mg aluminium alloys. The analysis of the results allowed the following conclusions to be drawn:Inclusions of about 10 µm in size existed at the bottom of the voids in the Al-7.5Mg sample (Figure 15a); the analysis also revealed the presence of Si, S, Ca and an elevated concentration of C and O. This analysis probably resulted from an inclusion that was formed during casting;The elements O and C registered on the spectrogram of the Al-3Mg sample (Figure 15b) were most likely a result of contamination of the fracture surface.Inclusions of about 4 µm in size existed at the bottom of the void on the fracture of the Al-1Mg sample (Figure 15c); the presence of Si and C suggests that it was an inclusion from a preliminary alloy.

The inclusions and precipitations identified affected the phenomena accompanying ductile fracture, which, as a result, was manifested by the size of the voids on the fracture surfaces. Depending on the chemical composition and type of heat treatment, Al-Mg alloys were strengthened by particles of intermetallic phases of the type: Mg_2_Al_3_, Al_6_(Mn, Fe), Al_6_Mn, Al_6_Fe.

## 4. Conclusions

The aim of the investigations presented in this article was to determine the effect of the addition of magnesium on the static and dynamic strength of Al-1Mg, Al-3Mg and Al-7.5Mg aluminium alloys. The results were compared with a reference material, i.e., Al99.5 aluminium. Moreover, the effect of the strain rate on the fracture behaviour was studied using fractographic analysis and energy dispersive X-ray spectroscopy. The research results allowed the following conclusions to be drawn:With an increase in magnesium content in the aluminium alloys tested under static conditions, an increase in the values of UTS and YS was observed. The same alloys strengthened in the HE process (logarithmic strain u = 0.86) showed an increase in YS of 85.6–277% and UTS by 45.7–80.5% in relation to the properties of Al99.5 aluminium. At the same time, the relative elongation value decreased by 85.7–95%;The content of 1% magnesium in Al-1Mg aluminium alloy caused a PLC effect called serration above the yield point and a change in the mechanical properties in relation to Al99.5 aluminium;For HE-strengthened aluminium alloys, an increase in the strain rate from 750 to 1750 s^−1^ caused an increase in dynamic UTS Rmd from 1.2 to 1.9 times in relation to static UTS;All samples fractured in the ductile fracture mode according to the classic mechanism: nucleation of the voids; growth of voids as a result of attaching new dislocations to them and then coalescence of voids leading to crack propagation;An increase in Mg content resulting in the formation of a larger amount of strengthening phase affected a different state of stress during dynamic loading, leading to a change in the orientation of the fracture surface;An increase in the content of Mg was associated with an increased number of voids, which number was additionally directly proportional to the strain rate in the dynamic tensile test.

Magnesium addition significantly influenced Al_2_O_3_ protective layer properties of Al-Mg alloys at an ambient temperature. Al_2_O_3_ also formed a protective layer that reduced the coefficient of friction of the material. Future studies should investigate the effect of the magnesium content on the corrosion resistance and tribological properties of the Al-Mg alloys after hydrostatic extrusion. An interesting research direction may be to analyse the protective nature of Al_2_O_3_ by increasing the electrochemical activity of the metal surface. Microstructural changes introduced to the Al-Mg alloys during severe plastic deformation include increased dislocation density, segregation of alloying elements grain refinement and presence of internal stresses. Therefore, the influence of residual stresses on the stress corrosion resistance will be the next topic of future works. Al-Mg alloys are used in the construction, chemical and shipbuilding industries. So, it would be interesting to investigate fatigue properties of the HEed specimens fabricated with a wide range of extrusion ratios and magnesium content.

## Figures and Tables

**Figure 1 materials-15-01066-f001:**
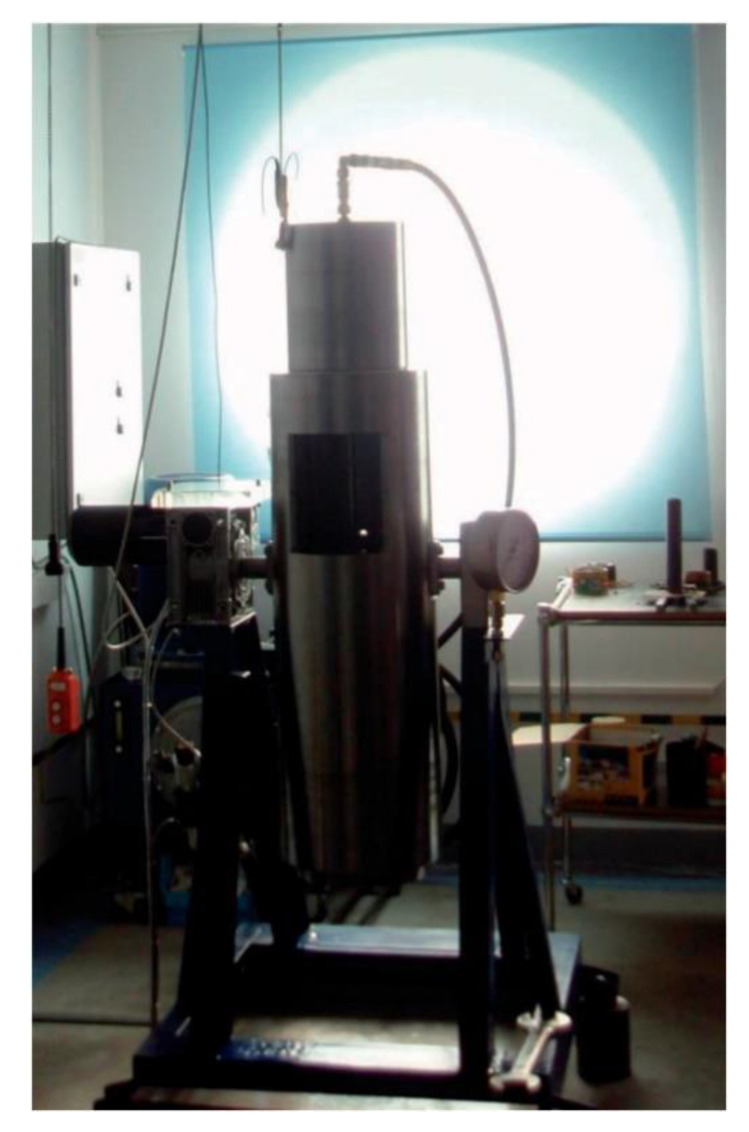
Hydrostatic extrusion press.

**Figure 2 materials-15-01066-f002:**
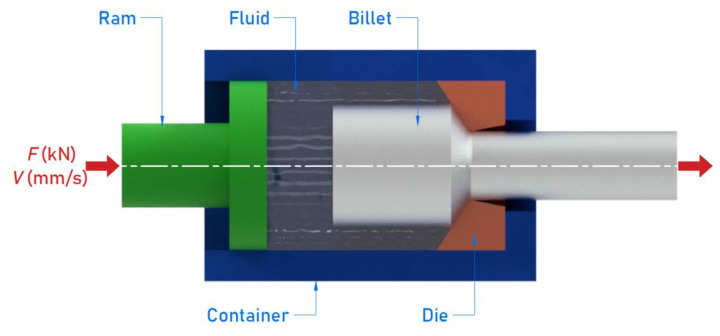
Schematic diagram of the extrusion tool.

**Figure 3 materials-15-01066-f003:**
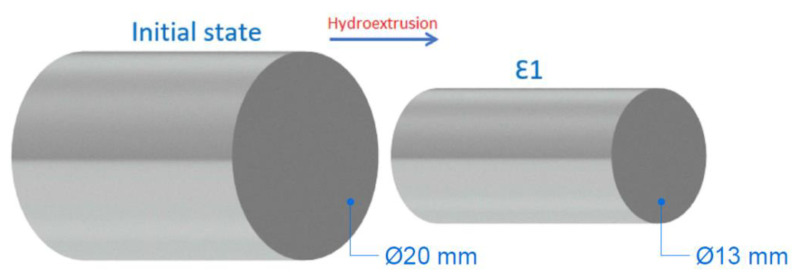
Schematic illustration of the deformation process.

**Figure 4 materials-15-01066-f004:**
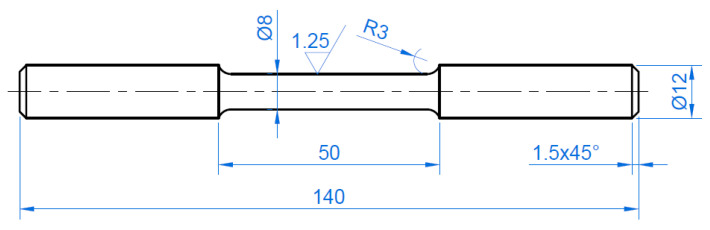
Shape and dimensions (in mm) of the specimens used in the uniaxial tensile tests.

**Figure 5 materials-15-01066-f005:**
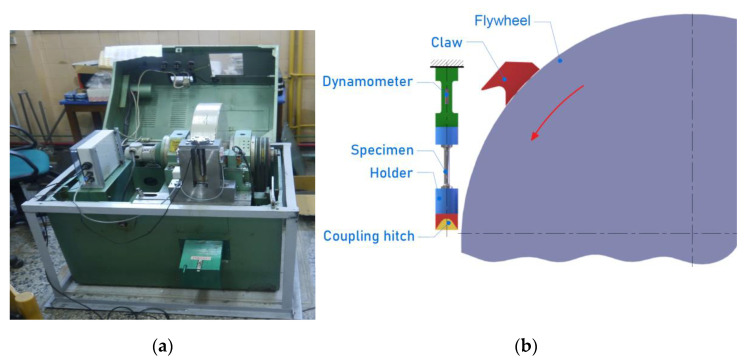
(**a**) Picture of rotary hammer and (**b**) schematic diagram of sample stretching using a rotary hammer.

**Figure 6 materials-15-01066-f006:**
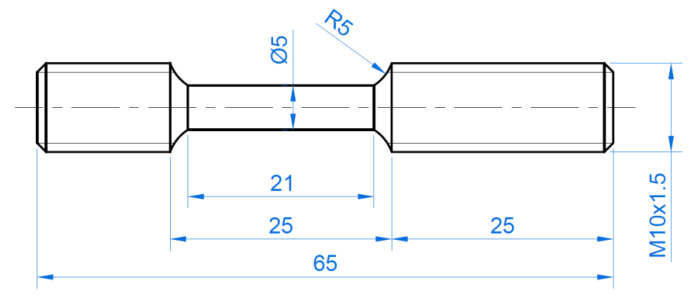
Shape and dimensions (in mm) of the specimens used in the dynamic tests.

**Figure 7 materials-15-01066-f007:**
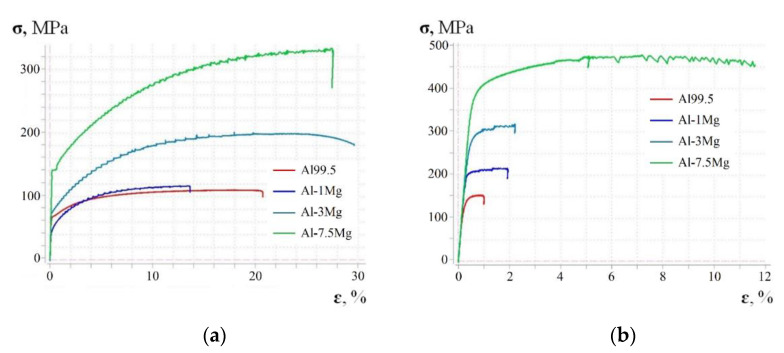
Selected (nominal) characteristics of the static tensile test of Al99.5 aluminum and Al-Mg alloys with variable Mg content: (**a**) Materials without strengthening with HE processing (u = 0), (**b**) materials after HE treatment (u = 0.86).

**Figure 8 materials-15-01066-f008:**
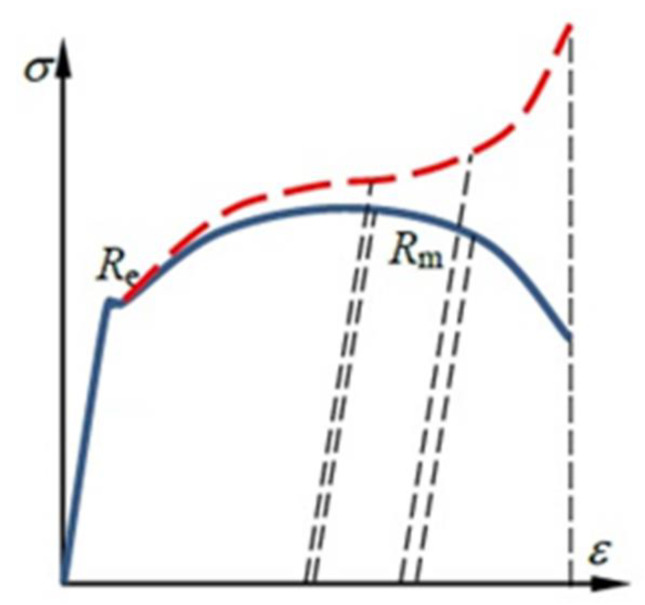
Nominal (blue line) and true stress–strain (red line) characteristic of material: R_e_—yield stress, R_m_—ultimate tensile stress [26].

**Figure 9 materials-15-01066-f009:**
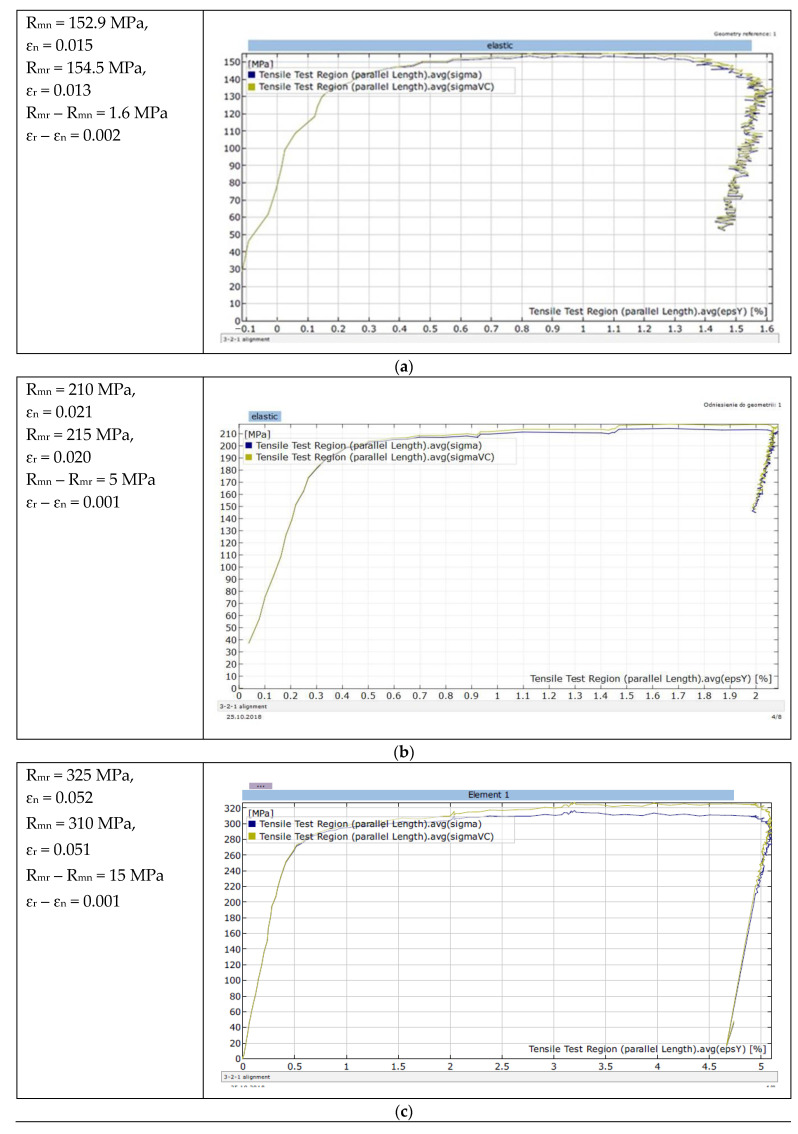
Selected nominal σ_nom_ = f(ε_nom_) (blue line) and true σ_true_ = f(ε_true_) (green line) curves for HEed specimens (u = 0.86): (**a**) Al99.5 aluminium, (**b**) Al-1Mg, (**c**) Al-3Mg and (**d**) Al-7.5Mg.

**Figure 10 materials-15-01066-f010:**
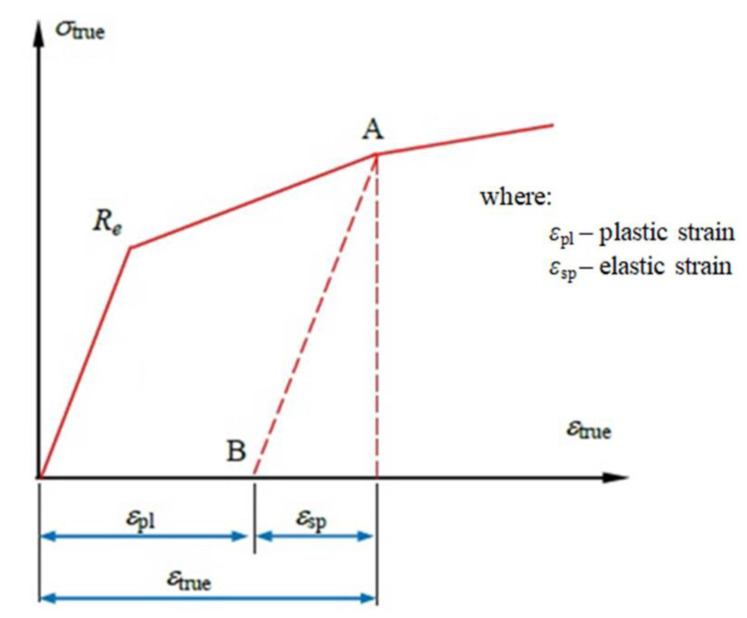
Division of the true strain ε_true_ into its elastic and plastic elements: R_e_—yield stress, A–B—unloading line. [27].

**Figure 11 materials-15-01066-f011:**
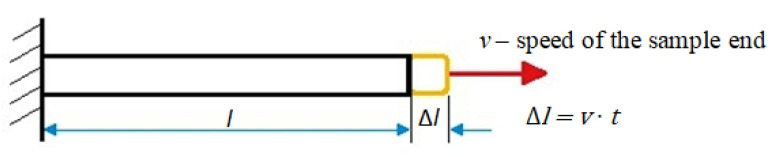
Sample deformation over time [25].

**Figure 12 materials-15-01066-f012:**
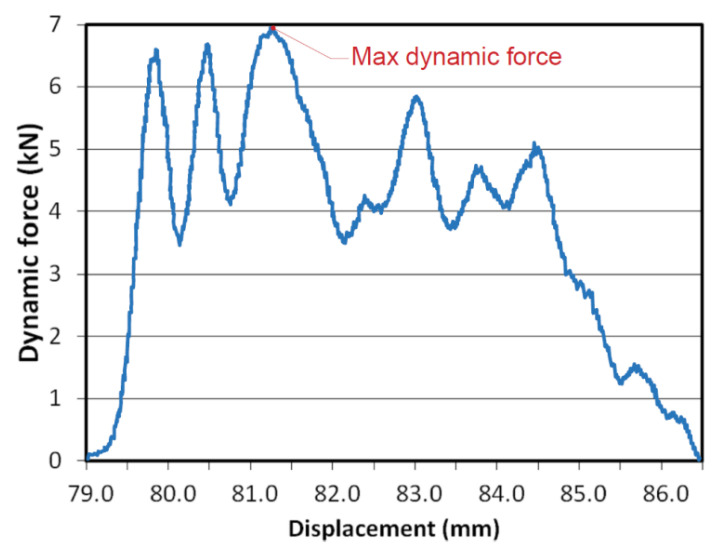
Record from the oscilloscope of the breaking force on a rotary hammer and the reading of the maximum dynamic force of a sample made of Al99.5 aluminium at a strain rate of 1250 s^−1^ (25m·s^−1^).

**Figure 13 materials-15-01066-f013:**
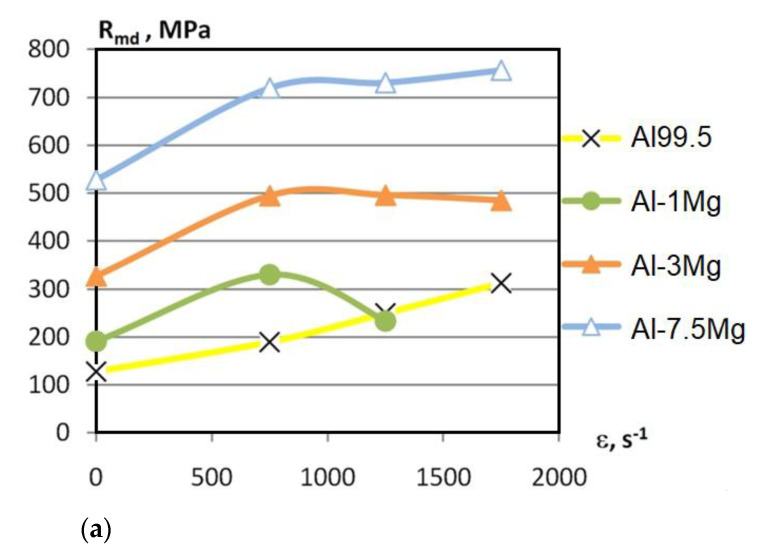
Variation of (**a**) dynamic strength R_md_ and (**b**) dynamic elongation A_5d_ depending on deformation speed ε.

**Figure 14 materials-15-01066-f014:**
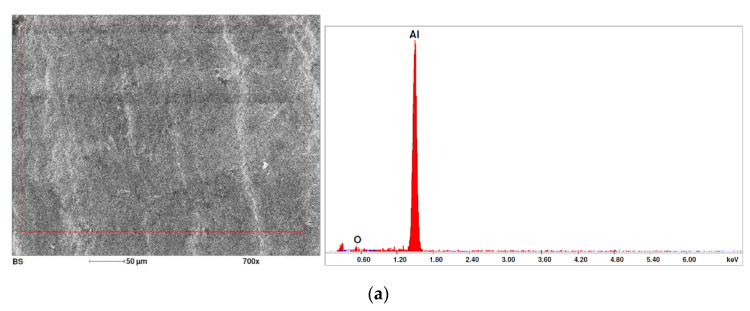
SEM micrographs (left) and EDS spectra (right) of side surface of samples for variants: (**a**) Al99.5, (**b**) Al-1Mg, (**c**) Al-3Mg and (**d**) Al-7.5Mg.

**Figure 15 materials-15-01066-f015:**
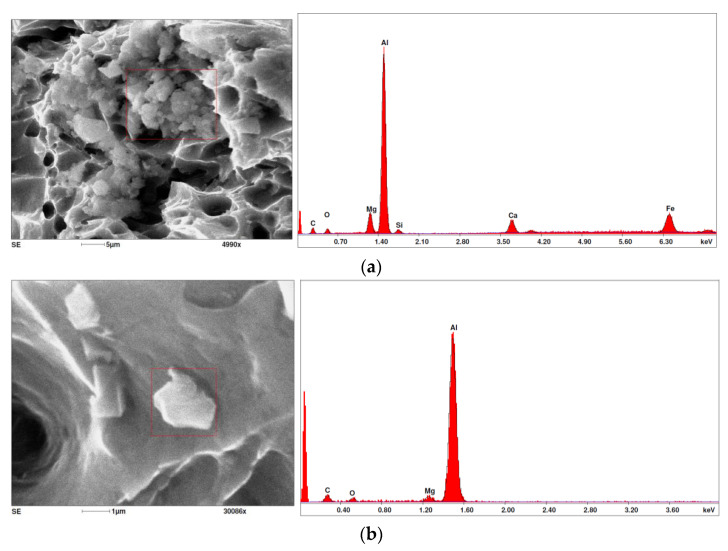
SEM micrographs (left) and EDS spectra (right) of inclusions in (**a**) Al-7.5Mg, (**b**) Al-3Mg and (**c**) Al-1Mg.

**Table 1 materials-15-01066-t001:** Chemical composition of the Al99.5 aluminium (wt.%).

Fe	Si	Cu	Zn	Ti	Mn	Mg	Ni	Sn	Pb	Al
0.097	0.04	0.0037	0.0038	0.006	0.0016	0.0007	0.0054	0.001	0.0048	99.82

**Table 2 materials-15-01066-t002:** Chemical composition of the Al-1Mg aluminium alloy (wt.%).

Fe	Si	Cu	Zn	Ti	Mn	Mg	Ni	Cr	Pb	Be	Al
0.087	0.040	0.0064	0.0036	0.0045	0.0015	1.016	0.0056	0.0009	0.0001	0.004	98.83

**Table 3 materials-15-01066-t003:** Chemical composition of the Al-3Mg aluminium alloy (wt.%).

Fe	Si	Cu	Zn	Ti	Mn	Mg	Ni	Cr	Pb	Be	Al
0.096	0.076	0.007	0.005	0.004	0.004	3.06	0.006	0.001	0.001	0.006	balance

**Table 4 materials-15-01066-t004:** Chemical composition of the Al-7.5Mg aluminium alloy (wt.%).

Fe	Si	Cu	Zn	Ti	Mn	Mg	Ni	Pb	Sn	Al
0.16	0.08	0.001	0.0001	0.016	0.006	7.52	0.002	0.003	0.0001	balance

**Table 5 materials-15-01066-t005:** True and nominal parameters of the material tested in the uniaxial tensile test.

Nominal Stress	Nominal Strain	True Strain	True Stress	Plastic Strain
$\sigma_{\mathrm{nom}}=\frac{F}{A_{0}}$	$\varepsilon_{\mathrm{nom}}=\frac{\Delta l}{l_{0}}$	$\varepsilon_{\mathrm{true}}=\ln\left(1+\varepsilon_{\mathrm{nom}}\right)$	$\sigma_{\mathrm{true}}=\sigma_{\mathrm{nom}}\times \left(1+\varepsilon_{\mathrm{nom}}\right)$	$\varepsilon_{\mathrm{pl}}=\varepsilon_{\mathrm{true}}-\frac{\sigma_{\mathrm{true}}}{E}$

**Table 6 materials-15-01066-t006:** List of the static mechanical properties of the materials tested, determined from the true (subscript r) and nominal (subscript n) characteristics for the degree of deformation of u = 0.86.

Material ϕ = 13 mm u = 0.86	Mechanical Properties							R_mr_ – R_mn_ (MPa)	
	True				Nominal				
	Yield Stress, R_p0,2r_, (MPa)	Strain ε_0,2r_, -	Ultimate Tensile Stress R_mr_, (MPa)	Strain ε_mr_, -	Yield Stress, R_p0,2n_, (MPa)	Ultimate Tensile Stress R_mn_, (MPa)	Strain ε_mn_ -	Determined Analytically	Determined Using DIC
Al99.5	115.3	0.0020	155.1	0.0080	120.62	153.8	0.01	1.6	1.8
Al.-1Mg	114.5	0.0015	217.8	0.0161	196.34	213.4	0.02	4.4	5
Al.-3Mg	149.8	0.0021	322.4	0.0218	275.51	306.8	0.023	15.6	15
Al.-7.5Mg	172.9	0.0022	514.2	0.1018	361.37	477.3	0.11	36.9	39

**Table 7 materials-15-01066-t007:** Fracture morphology of samples of Al99.5 aluminium sample.

$\dot{\varepsilon }$, ${\text{s}}^{-1}$	View of Tensile Specimen after Testing	View of Fracture Surface	Fracture Morphology
750	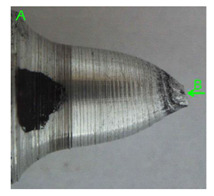	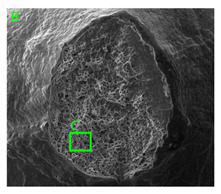	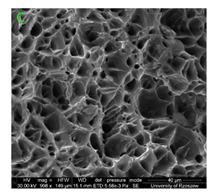
1250	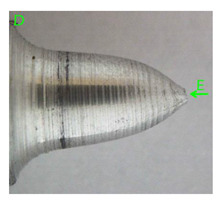	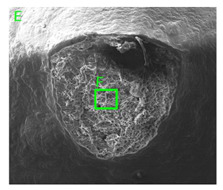	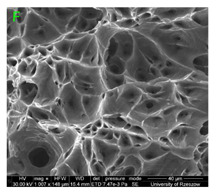

**Table 8 materials-15-01066-t008:** Fracture morphology of Al-1Mg sample.

$\dot{\varepsilon }$, ${\text{s}}^{-1}$	View of Tensile Specimen after Testing	View of Fracture Surface	Fracture Morphology
750	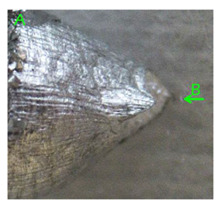	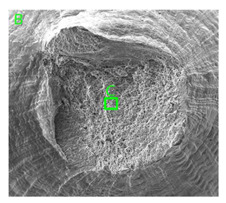	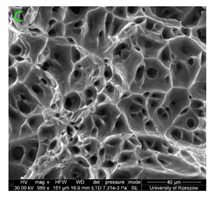
1250	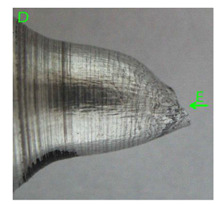	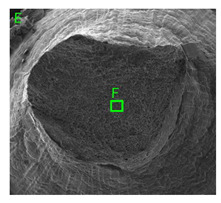	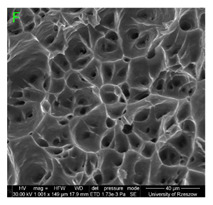

**Table 9 materials-15-01066-t009:** Fracture morphology of Al-3Mg sample.

$\dot{\varepsilon }$, ${\text{s}}^{-1}$	View of Tensile Specimen after Testing	View of Fracture Surface	Fracture Morphology
750	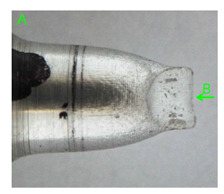	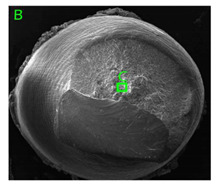	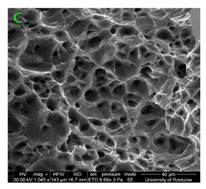
1250	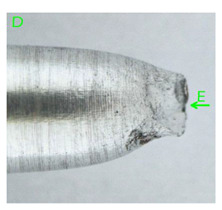	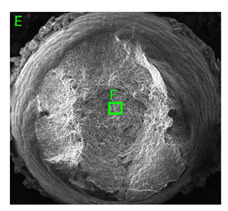	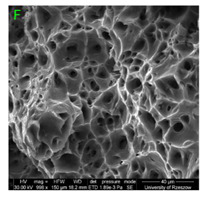

**Table 10 materials-15-01066-t010:** Fracture morphology of Al-7.5Mg sample.

$\dot{\varepsilon }$, ${\text{s}}^{-1}$	View of Tensile Specimen after Testing	View of Fracture Surface	Fracture Morphology
750	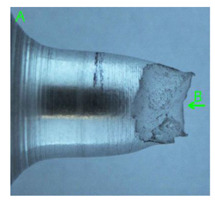	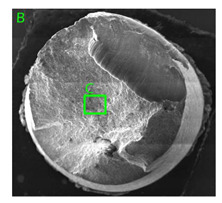	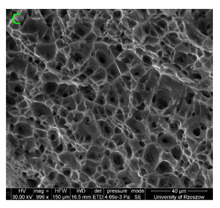
1250	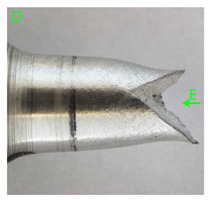	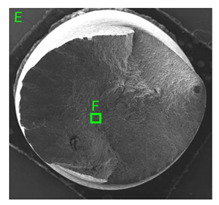	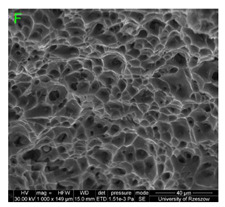

**Table 11 materials-15-01066-t011:** Results of a quantitative analysis of the voids in the fracture surface of Al99.5 aluminium sample.

$\dot{\varepsilon }$, ${\text{s}}^{-1}$	Boundaries of the Voids	Histogram
750	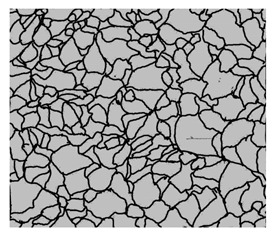	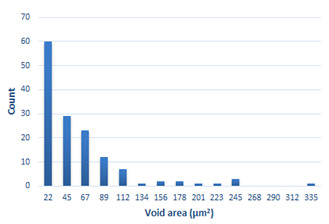
1250	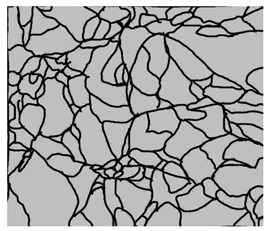	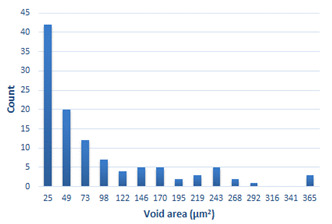
1750	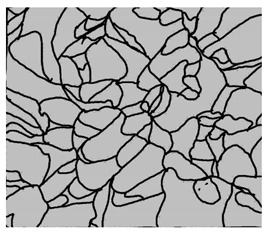	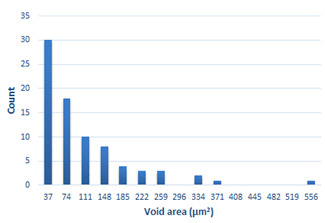

**Table 12 materials-15-01066-t012:** Results of a quantitative analysis of the voids in the fracture surface of Al-1Mg.

$\dot{\varepsilon }$, ${\text{s}}^{-1}$	Boundaries of the Voids	Histogram
750	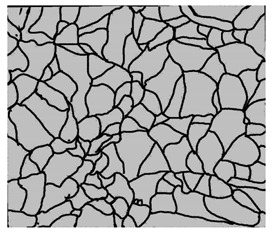	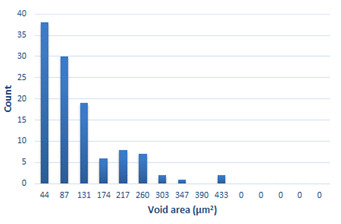
1250	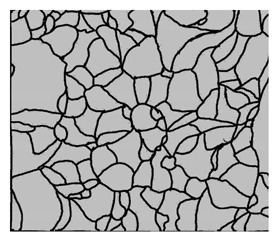	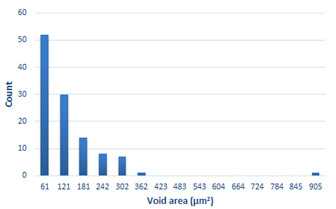

**Table 13 materials-15-01066-t013:** Results of a quantitative analysis of the voids in the fracture surface of Al-3Mg.

$\dot{\varepsilon }$, ${\text{s}}^{-1}$	Boundaries of the Voids	Histogram
750	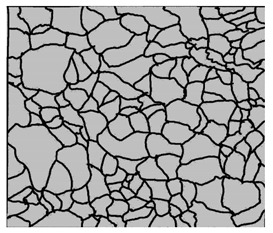	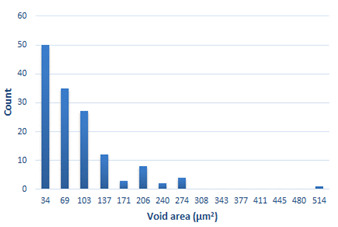
1250	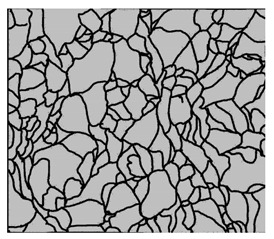	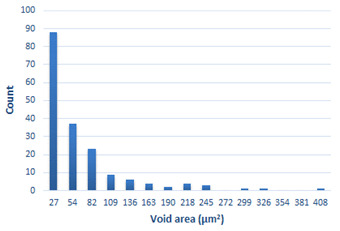
1750	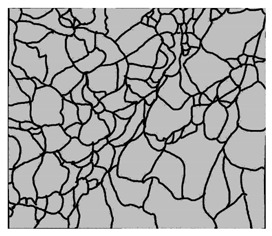	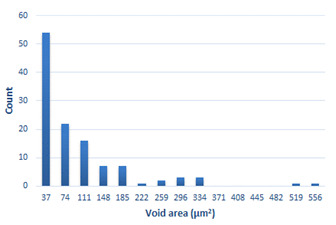

**Table 14 materials-15-01066-t014:** Results of a quantitative analysis of the voids in the fracture surface of Al-7.5Mg.

$\dot{\varepsilon }$, ${\text{s}}^{-1}$	Boundaries of the Voids	Histogram
750	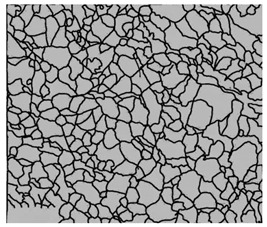	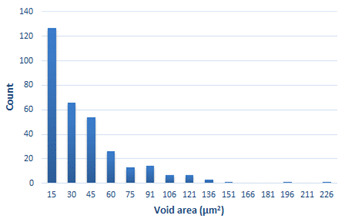
1250	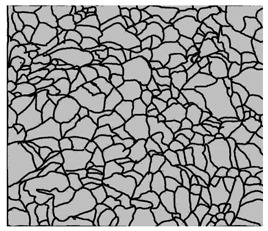	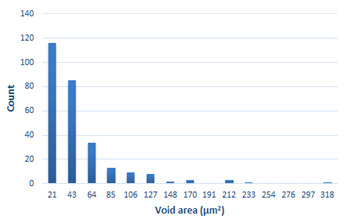
1750	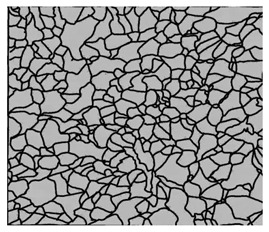	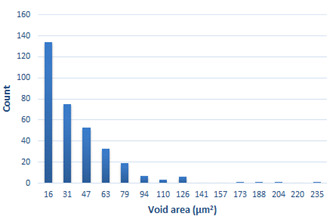

**Table 15 materials-15-01066-t015:** Summary of the quantitative analysis of voids in fracture surfaces.

Specimen	Number of Voids Per mm^2^	$\mathrm{Strain}\;\mathrm{Rate}\;\dot{\varepsilon }$ (s^−1^)
Al99.5_1	8913	750
Al99.5_2	6967	1250
Al99.5_3	5021	1750
Al.-1Mg_1	7093	750
Al.-1Mg_2	7093	1250
Al.-3Mg_1	8913	750
Al.-3Mg_2	11,236	1250
Al.-3Mg_3	7344	1750
Al.-7.5Mg_1	20,087	750
Al.-7.5Mg_2	17,263	1250
Al.-7.5Mg_3	20,966	1750

## Data Availability

The data presented in this study are available upon request from the corresponding author.

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
