# Peer review of "Static and Dynamic Properties of Al-Mg Alloys Subjected to Hydrostatic Extrusion"

_materials, 2022, doi:10.3390/ma15031066_

Round 1
Reviewer 1 Report
What is new being presented in this article? Put this information in the abstract.
The gap area in the research is not clear. How the present work is different from other researcher’s work should be mentioned clearly.
At the end of Introduction section, it would be better to add the paper's organization in different sections.
The work literature is very interesting, it is worth showing yet some selected paper from this range, for example:
https://doi.org/10.1515/corrrev-2020-0048
Wherever applicable, the scientific explanation needs to be added and the research novelties need to be clearly emphasized.
Some figures scale bars are missing.
Future work is missing.
Please check the manuscript for wrong choice of words, grammatical errors and incoherent sentence structure.
Author Response
Thank you for your valuable comments, we have marked all changes to the text in green. Below we present the answers to individual comments:
1. What is new being presented in this article? Put this information in the abstract.
Response: Additional information has been added in the „Abstract” section:
The aim of this study is to determine the influence of the amount of magnesium in Al-Mg alloys and strain rate on the grain refinement and mechanical properties of material determined in the dynamic tensile test. The hydrostatic extrusion was used to processing the material. This method is not commonly used method of impose severe plastic deformation of Al-Mg alloys.
2. The gap area in the research is not clear. How the present work is different from other researcher’s work should be mentioned clearly.
Response: Additional information has been added in the „Introduction” section:
Most published researchers investigated the effectiveness of the plastic deformation methods in improving the strength of Al-Mg alloys. Works were focused on the correlation between the amount of magnesium, deformation degree and strengthening of the processed materials. Many of them indicated also that large plastic deformations resulted in the improvement of properties such as microhardness without deteriorating properties such as and electrical conductivity and ductility. The aim of this study is to determine the influence of the amount of magnesium and strain rate on the grain refinement and mechanical properties of material determined in the dynamic tensile test. Moreover, in this paper the hydrostatic extrusion is used to processing the material. This process requires special equipment and is not commonly used method of impose severe plastic deformation of Al-Mg alloys.
3. At the end of Introduction section, it would be better to add the paper's organization in different sections.
Response: Additional information has been added in the „Introduction” section:
This paper is organized as follows. The research methodologies adopted to characterize and assess the properties of the Al-Mg alloys are presented in Section 2. The results of the static and dynamic tensile tests are presented in Sections 3.1. and 3.2, respectively. Analysis of the fracture morphology of samples after dynamic test is shown in Section 3.3. Towards to the end of this article analysis of voids on the fracture surfaces (Section 3.4) and sources of the nucleation of voids (Section 3.5) are discussed and finally conclusions and future work plan are made.
4. The work literature is very interesting, it is worth showing yet some selected paper from this range, for example:
https://doi.org/10.1515/corrrev-2020-0048
Response: Additional information has been added in the „Introduction” section:
Aluminium metal matrix composites (AMMCs) fabricated using powder metallurgy are expanding their range due to their favour characteristics sych as high-temperature resistance, high specific strength and light weight than conventional materials. Behera et al. [6] fabricated aluminum-metal–matrix-composite (Al–0.5Si–0.5Mg–2.5Cu–5SiC). They found that incorporation of fine SiC particles into sintered matrix element can improve erosive wear resistance, by a factor of 200–300%. Form the observations of many authors (Abbas et al. [7], Behera et al. [8], El-Aziz et al. [9]), it is noticed that AMMCs are capable to improve the erosion behaviors and degradation of corrosion. Behera et al. [10] found that the hardness of sintered AMMC is increased with increasing the amount of Si-C reinforcements.
5. Wherever applicable, the scientific explanation needs to be added and the research novelties need to be clearly emphasized.
Response: The manuscript has been updated. Please see our resonses to you comments no. 1 and 2.
6. Future work is missing.
Response: Thank you for your significant comments. The following except has been added in the “Conclusions” section:
Magnesium addition significantly influences Al2O3 protective layer properties of Al-Mg alloys in an ambient temperature. Al2O3 also forms a protective layer that reduces the coefficient of friction of the material. Future studies should investigate the effect of the magnesium content on the corrosion resistance and tribological properties of the Al-Mg alloys after hydrostatic extrusion. An interesting research direction may be to analyse the protective nature of Al2O3 by increasing the electrochemical activity of the metal surface. Microstructural changes introduced to the Al-Mg alloys during severe plastic deformation include increased dislocation density, segregation of alloying elements grain refinement and presence of internal stresses. Therefore, the influence of residual stresses on the stress corrosion resistance will be the next topic of future works. Al-Mg alloys are used in the construction, chemical and shipbuilding industries. So it would be interesting to investigate fatigue properties of the HEed specimens prepared for wide range of extrusion ratios and magnesium content.
7. Please check the manuscript for wrong choice of words, grammatical errors and incoherent sentence structure.
Response: The minor mistakes have been corected in the manuscript. Please note, that the manuscript (IF ACCEPTED) is obligatory proofread by publisher.
Reviewer 2 Report
The current paper studies the effect of adding different Mg to pure Al alloy on the Static and dynamic properties. The pure Al alloy and Al-xMg (x=1, 3, and 7.5wt.%) alloys were casted and subsequently homogenized. The homogenized pars were hydrostatic extruded, then the Static and dynamic properties were investigated. The current paper is worthy of investigation and organization. The following comments need to be addressed to be accepted for publication.
- The proficiency of the language needs more improvement in the manuscript
- In the abstract, I have confused by AlMg1 AlMg3 and AlMg7.5. the numbers 1, 3, and 7.5 are the wt%. so, I recommend writing it as Al-xMg (x=1, 3, and 7.5wt.%).
- More details are required about the casting and homogenization processes of the investigating materials.
- The unit of the dimensions in Figure 3 should be added.
- Authors didn’t mention in the dynamic tensile test how many samples were tested. And which standard you used?
- Please unify the schematic drawing of the used samples in static and dynamic tensile tests. (lines, and dimensions style)
- The SEM and EDS spectra of the as-extruded materials should be added.
- The authors should state how the quantitative analysis of the voids on the fracture surface was performed. Regarding Tables 11-14, the x-axis of the histograms, what is the meaning of size of voids-number per mm2? Is it mean the planar density (voids/mm2)? If that, so it isn't size.
- Figure 7, High quality is required.
- Regarding Figure 12, and Figure 13, please used one style for all curves.
Author Response
Thank you for your valuable comments, we have marked all changes to the text in yellow. Below we present the answers to individual comments:
· The proficiency of the language needs more improvement in the manuscript
Response: The minor mistakes have been corected in the manuscript. Please note, that the manuscript (IF ACCEPTED) is obligatory proofread by publisher
· In the abstract, I have confused by AlMg1 AlMg3 and AlMg7.5. the numbers 1, 3, and 7.5 are the wt%. so, I recommend writing it as Al-xMg (x=1, 3, and 7.5wt.%).
Response: Thank you very much for this remark, we initially used such markings during the research, but in fact they may be ambiguous for the reader and may be misleading. We therefore revised the markings throughout the manuscript as suggested.
· More details are required about the casting and homogenization processes of the investigating materials.
Response:
Yes, of course the scientific articles should provide all the details of the sample preparation, but please understand that this is an exceptional case. The tested materials were produced by the Institute listed below, and the details of their production, including the casting and homogenization processes, are the secret of this Institute and cannot be disclosed to the public.
Institute of Non-Ferrous Metals - Łukasiewicz Research Network
Light Metals Division
ul. Pilsudskiego 19
32-050 SKAWINA
Poland
Such information about the secrecy of the Institute was included in the text of the manuscript.
· The unit of the dimensions in Figure 3 should be added.
Response: Units have been introduced in Figure 3.
· Authors didn’t mention in the dynamic tensile test how many samples were tested. And which standard you used?
Response: Three specimens were tested for each material, such information is provided in the text. The dynamic tests were carried out according to the recommendations of the rotary hammer manual.
· Please unify the schematic drawing of the used samples in static and dynamic tensile tests. (lines, and dimensions style)
Response: Figures 4 and 6 have been standardized in terms of style.
· The SEM and EDS spectra of the as-extruded materials should be added.
Response: Thank you for such valuable attention, we have added SEM images and EDS analysis results as Figure 14 in the text.
· The authors should state how the quantitative analysis of the voids on the fracture surface was performed. Regarding Tables 11-14, the x-axis of the histograms, what is the meaning of size of voids-number per mm2? Is it mean the planar density (voids/mm2)? If that, so it isn't size.
Response: In the tested samples, heterogeneity in the structure of the fracture was observed, manifested by a different size and unequal shape of the wells. The shape of the wells, which depends on the local loading conditions, shows the majority of normal stresses.
Fractographic quantitative tests were made on the basis of SEM photographs of the fracture surface made at a magnification of 1000x. The images were analyzed using dedicated quantitative image analysis software - Multiscan. After using morphological filters, wells were outlined and the area of each void was measured in square micrometers µm2. From the obtained results, histograms were made: number of voids - void surface area (determined by the range of numbers).
The large number of voids per 1mm2 indicates a limited role of the mechanism of nucleation of micro-voids and the joining of micro-voids in these alloys in the process of creating the fracture surface.
In reference to the histograms, the axes were incorrectly described earlier due to our error. Currently we have changed the axis signatures, at the moment there is 'Void area (µm2)' on the x-axis and 'Count' on the y-axis, which more precisely indicates the described phenomena
· Figure 7, High quality is required.
Response: Better quality of Figure 7 has been introduced, captions have been changed to be more readable. Thank you for your attention, in fact the charts were previously unreadable.
· Regarding Figure 12, and Figure 13, please used one style for all curves.
Response: Figure 12 has been adapted to the style of the plots in Figure 13.
Round 2
Reviewer 1 Report
Authors have made significant changes in the revised manuscript. Hence, accept the manuscript for publication in its present form.
Reviewer 2 Report
The authors have addressed the all comments. The paper is recommended to be accepted in the current form.